# Long-Term Survival of Patients with Adult T-Cell Leukemia/Lymphoma Treated with Amplified Natural Killer Cell Therapy

**DOI:** 10.3390/reports7030080

**Published:** 2024-09-19

**Authors:** Yuji Okubo, Sho Nagai, Yuta Katayama, Kunihiro Kitamura, Kazuhisa Hiwaki, Keisuke Teshigawara

**Affiliations:** 1Higashinotohin Clinic, 338 Empukuji, Nakagyo, Kyoto 604-8175, Japan; yjokb61@gmail.com; 2Ebino Centro Clinic, 1007-4, Uwae, Ebino, Miyazaki 889-4304, Japan; ebino280centro@yahoo.co.jp; 3Hiroshima Red Cross Hospital & Atomic-bomb Survivors Hospital, 1-9-6, Senda, Naka, Hiroshima 730-8619, Japan; yutak1022@nifty.com; 4Kitamura Clinic, 4-3-8, Nishiki, Onojo, Fukuoka 816-0935, Japan; kunihiro@kitamura.or.jp; 5Hiwaki Clinic, 2-14-65, Asano, Kokurakita, Kitakyushu, Fukuoka 802-0001, Japan; hiwakiclinic@yahoo.co.jp

**Keywords:** ATL, HTLV-1, sIL-2R, NK cells, ANK therapy, oncology

## Abstract

Background: Adult T-cell leukemia/lymphoma (ATL) is caused by human T-cell leukemia virus type 1 (HTLV-1) after a long latent infection. HTLV-1 induces the indolent or aggressive type of leukemia in 5% of HTLV-1 carriers. ATL, especially the aggressive type, is resistant to multi-agent chemotherapy. The indolent type often progresses to the aggressive type. Even in the most indolent-type cases, that is, smoldering ATL, the average survival time is 55.0 months. Case Presentation: Five patients with ATL were followed up for their clinical course after amplified natural killer cell (ANK) therapy. Four patients who received ANK therapy as first-line therapy achieved complete remission and showed long-term survival without aggressive conversion or relapse for more than 5 years. One patient was treated with multiagent chemotherapy due to acute exacerbation but relapsed 2 months later. She was subsequently treated with radiation and ANK therapy and survived for more than 6 years. Furthermore, ANK therapy enhanced the immune function of ATL patients to a level higher than that of normal individuals. Conclusions: ANK therapy has great potential as first-line treatment for ATL.

## 1. Introduction

Adult T-cell leukemia/lymphoma (ATL) is an aggressive lymphoproliferative disease caused by human T-cell leukemia virus type 1 (HTLV-1) infection [1,2]. There is a long latency period between HTLV-1 infection and the onset of clinical ATL, which usually occurs in older patients over 40 years of age [3,4,5]. Only a small proportion of HTLV-1 infected carriers—6% of men and 2% of women—develop ATL [4]. ATL is classified into four clinical types: acute, lymphoma, chronic, and smoldering [4,5]. Acute and lymphoma types have a poorer prognosis because of their rapid progression, frequent recurrence, and severe immunosuppression [4,5,6]. Therefore, they are categorized as aggressive. In contrast, chronic and smoldering ATLs are categorized as indolent because their prognosis varies widely among individuals [4].

Patients with ATL have limited treatment options because of the intrinsic resistance of ATL to conventional chemotherapy [4]. The aggressive type (acute and lymphoma) often shows acute deterioration and relapse shortly after the induction of complete remission (CR) with multi-agent chemotherapy [7]. In the indolent (chronic and smoldering) type, even if obvious hematologic abnormalities are observed, “watchful waiting” is usually recommended in Japan unless unfavorable prognostic factors, such as elevated LDH and BUN and decreased albumin, appear.

In ATL and non-Hodgkin’s lymphoma, elevated serum levels of soluble interleukin-2 receptor (sIL-2R) reflect poor prognosis [8,9]. Therefore, the efficacy of treatment for ATL, such as remission, exacerbation (conversion from the indolent to the aggressive type), or relapse, can be evaluated using serum sIL-2R levels [6,10]. Katsuya et al. analyzed the prognosis of both aggressive and indolent ATLs and reclassified them into high-, intermediate-, and low-risk groups, primarily according to serum sIL-2R levels [8,9]. The 2-year overall survival (OS) rates for the aggressive form are 4, 17, and 39% for the high-, intermediate-, and low-risk groups, respectively. For the indolent type, the 4-year OS rates for the high-, intermediate-, and low-risk groups are 26.2, 55.6, and 77.6%, respectively [8,9]. The prognosis for the aggressive type is very poor in both the high- and intermediate-risk groups. Even the indolent type, as seen in Patient 1 in this study, generally has a poor prognosis once converted into the aggressive type. Skin involvement occurs in nearly half of patients with ATL, and the nodulotumoral and erythrodermic subtypes are associated with poor prognoses [11]. For skin lesions of indolent ATL, patients are treated with topical skin therapy, including corticosteroids and retinoids, local radiation therapy, and photochemotherapy. To prevent the acute exacerbation of ATL, the therapy includes the systemic administration of steroid hormones, oral retinoids, interferon (IFN)-γ preparations, and/or VP-16. However, the effects of these therapies are not long-lasting, as the average survival time for the chronic and smoldering types is 31.5 and 55.0 months, respectively. There is no evidence that these therapies contribute to improved survival [12].

Novel first-line therapies are needed to provide long-term clinical benefits to patients with ATL. ANK therapy, in which natural killer (NK) cells are harvested from cancer-bearing patients and then activated, proliferatively amplified, and autologously administered, has been previously reported [13]. ANK therapy specifically attacks tumor cells, thus posing little risk of serious damage to the normal immune system. Here, we describe our experience with ANK therapy for various types of ATL diagnosed according to the ATL Prognostic Index Project [8,9].

## 2. Case Presentation Section

This study presents the clinical course of five patients who were followed for more than five years after ANK therapy (Table 1, Figure 1).

ANK therapy was performed as follows: briefly, peripheral blood mononuclear cells (PBMCs) were isolated from the whole blood of each patient using Ficoll density gradient centrifugation (Lymphoprep; Alere Technologies, Oslo, Norway) or leukapheresis. The PBMCs (1 × 10^6^ to 5 × 10^6^/mL) were activated and cultured in a medium supplemented with IL-2 and IL-15 [13]. Half of the culture medium was replaced with a fresh medium every 2–3 days. After 2 to 3 weeks, the surface marker analysis demonstrated that 50–90% of the cultured lymphocytes were CD3-negative and CD56-positive, thus indicating that the majority had an NK cell phenotype. Since high NK cell activity was confirmed by cytotoxic activity targeting K562 cells even after culture, these cultured cells were termed ANK cells. ANK cells were aliquoted into multiple cryotubes and stored at −80 °C until use. Prior to treatment, frozen ANK cells were thawed and cultured in fresh media. The viability of the thawed cells ranged from 70% to 90% depending on the condition of the patient from whom the blood was drawn but increased to over 95% after one week of re-culture. We observed decreased NK cell activity 3 to 4 days after a single injection of ANK cells. Therefore, we performed ANK therapy twice a week to maintain NK cell activity in vivo. ANK cells produce various cytokines, including IFN-γ. The reason for the limited infusion of cultured ANK cells is to prevent the development of cytokine release syndrome; thus, 2–5 × 10^8^ cells were administered per injection [13].

The patients in this study were categorized according to clinical type and risk factors (mainly serum sIL-2R). Patients 1 and 2 were in the high-risk group. Patients 3, 4, and 5 were in the low-risk group [8,9]. Patient 1 was treated with ANK therapy upon the exacerbation of smoldering ATL, as previously reported [13]. Patient 2 was initially diagnosed with cutaneous T-cell lymphoma and then diagnosed with cutaneous ATL 5 months later after the detection of HTLV-1 virus. During the first 6 years after their initial diagnosis, Patient 2 received single-agent chemotherapy (VP-16) to prevent acute exacerbation. One skin tumor appeared on the right buttock at 5 years and 11 months after the diagnosis. Then, 6 years and 3 months later, multiple skin tumors appeared; 7 years later, the serum sIL-2R levels rose to 119,981 U/mL, indicating acute exacerbation of ATL (nodulotumor type, Figure 2).

The patient had been treated with mogamulizumab, but due to its severe side effects, she discontinued the treatment and switched to multi-agent chemotherapy.

For Patients 3–5, ANK therapy was attempted to decrease the risk of acute exacerbation and analyze its long-term clinical effects (Table 1).

For Patient 1, at the time of exacerbation, serum sIL-2R levels increased to 11,000 U/mL, but after two courses of ANK therapy (11 and 4 doses), the levels decreased to less than 2000 U/mL, resulting in CR. Thereafter, she did not need any supportive therapy and maintained CR for 5 years and 6 months after the start of ANK therapy until she died of myocardial infarction (Figure 3).

Being so effective as first-line therapy for Patient 1, ANK therapy was attempted as first-line therapy in Patients 3, 4, and 5 to decrease the risk of conversion to the aggressive type. These patients achieved CR, and their lymphadenopathy and skin lesions disappeared. These skin lesions are induced by ATL producing proinflammatory cytokines and/or ATL invasion. In particular, skin lesions of the nodulotumoral and erythrodermic types are derived from ATL invasion. In Patients 3 and 4, who were treated with ANK therapy, serum sIL-2R levels increased temporarily however returned to and remained at normal levels (Figure 4).

Patient 2 developed an aggressive type of ATL with multiple tumors. CR was induced using multi-agent chemotherapy; however, ATL recurred, and serum sIL-2R levels increased (nodulotumor type, Figure 5).

In general, the prognosis for both the acute and lymphoma types is poor, but it is even worse for these types with multiple skin tumors.

Since Patient 2 had already received multi-agent chemotherapy, she underwent repeated single-agent chemotherapy and radiation on the skin tumors. Radiation on the skin tumors, except those on the left forearm, was performed. ANK therapy was applied between each course of radiation. Radiation and ANK therapy decreased serum sIL-2R levels to 2655 U/mL after the fifth course (Figure 5). ANK therapy was able to eliminate multiple skin tumors on the left forearm, even though no radiation was delivered there (Figure 6, red arrows; Photo 2).

Interestingly, serum sIL-2R levels decreased after the seventh course of ANK therapy, even though previous radiotherapy failed to reduce the serum sIL-2R levels (Figure 5, black star sign). For Patient 2, ANK therapy was discontinued because of her bone fracture.

To avoid cytokine release syndrome, the dose of the ANK cells used was limited, but some patients experienced a temporary high fever, which improved by the next day and returned to healthy levels. In four patients who received ANK therapy (Patients 1, 3, 4, and 5), their CR status was maintained for more than 5 years, suggesting that ANK therapy enhanced their host immune function. NK cell activity in two CR patients (Patients 4 and 5) who survived for more than 7 years after ANK therapy was assayed using europium-labeled K562 cells as a target [14]. Interestingly, these patients showed higher NK cell activity compared with healthy controls (Figure 7).

## 3. Discussion

The pathophysiology and treatment of ATL were discussed and reported at an international meeting in 2018 [15]. However, to date, no decisive therapy exists. Despite various therapeutic trials, the prognosis for patients with ATL remains poor. Dose-enhanced multi-agent chemotherapy, called VCAP-AMP-VECP, has been recommended as first-line therapy for patients with aggressive ATL (<70 years, transplantation-eligible) [16]. The 2-year OS rate is reported to be 31.2% in the high-risk group; that of conventional chemotherapy (CHOP) is 24.6% and those of VCAP-AMP-VECP and CHOP are 39.8 and 45.0%, respectively, in the low-risk group [17]. Therefore, a better alternative therapy is desired. We previously reported a case of the smoldering ATL in which ANK therapy induced CR in acute crisis and maintained CR for 5 years [13]. We subsequently treated five patients with ATL who had ANK therapy, four of whom could be followed up for more than 5 years. The target of ANK therapy is specific to tumor cells, and the risk of severe damage to the immune system is extremely low. In this study, we applied ANK therapy as first- or second-line therapy for ATL. Among the patients who received ANK therapy as first-line therapy in this study, one showed acute exacerbation prior to treatment, as reported previously [13], while the other patients were under watchful waiting. CR was induced in all patients who received first-line ANK therapy and was maintained for more than 5 years without any supportive care.

As in Patient 2, in cases in which ANK therapy is used as second-line treatment, multi-agent chemotherapy may cause some damage to the immune system or some changes in tumor cells. Due to the limited number of ANK cells compared with ATL cells proliferating at the time of recurrence, the number of ANK cells might not be sufficient to kill all tumor cells. In two patients who achieved CR, the serum sIL-2R levels temporarily increased but then decreased without further treatment. The possibility that these increases were a sign of infection or ATL recurrence cannot be ruled out, but the cause was unclear since the patients were able to maintain CR status for more than 8 years. Careful future observation is needed to determine whether this surge was related to the success of ANK therapy. Interestingly, the two surviving patients who maintained CR more than 8 years after ANK therapy showed higher NK cell activity compared with healthy controls, suggesting that the immune system is critical in preventing the progression of ATL. The mechanism of efficacy of ANK therapy is considered to maintain high NK cell activity by inducing a memory-like response in cancer, which thereby kills the remaining ATL cells after CR and prevents the transformation of HTLV-1-infected cells [18]. Moreover, memory-like NK cells exemplify a highly effective approach for enhancing NK cell responses against cancer. After their discovery in mice in 2009 and humans in 2012, this innate cellular therapy was rapidly translated into clinical application, with AML patients treated with memory-like NK cells in 2014 [18,19]. Patient 3 died of lung cancer, even with low levels of serum sIL-2R, suggesting no relation to ATL. There is a possibility that activated NK cells stimulate the resting host immune system in the body through direct cell–cell interactions and/or their production of cytokines, such as IFN-γ and IL-2 [20], especially in hematological and/or virus-related malignancies.

ATL usually develops in old age, suggesting that age-related decline in immune function is associated with the development of ATL. ATL cells have the characteristics of regulatory T-cells [21] and express inhibitory ligands, such as PD-L1 [22], giving negative signals to T-cells and producing IL-10 and TGF-β, which suppress the immune response [23]. Furthermore, HTLV-1-infected cells produce proinflammatory cytokines, such as IL-6, IL-13, and IL-17, which induce chronic inflammation [24,25]. IL-10 and IL-17 suppress the enhanced immune response induced by IFN-γ and are associated with skin inflammation. The cytokine network in patients with ATL is dysregulated [26] and the induced chronic inflammation increases the risk of oncogenesis [26,27]. Clinical data show that IFN-γ levels are decreased in patients with ATL [28]. In general, the prognosis for both the acute and lymphoma types is poor and worse for those with multiple skin tumors. Skin lesions of ATL usually recur, regardless of the application of single-agent chemotherapy, radiation therapy, or corticosteroid hormones. The skin lesions and lymphadenopathy in Patients 4 and 5 disappeared with ANK therapy, and these lesions did not relapse.

It is important to develop effective therapies that can improve prognosis by eliminating immunosuppression by appropriately controlling cytokine networks in patients with ATL. Compared with T cells, activated NK cells are less sensitive to immunoinhibitory signals from cancer cells and efficiently kill cancer cells, irrespective of the PD-L1 expression level of cancer cells [29]. MHC Class I receptors also inhibit the cytotoxic activity of NK cells. However, activated NK cells express immune-activating receptors, including the NKG2D receptor, and recognize its ligands on cancer cells induced by stress or transformation. Even if cancer cells express a variety of molecules for immune escape, activated NK cells can kill cancer cells by inhibiting their activity via receptors for their ligands [30]. Since ANK therapy using ex vivo activated NK cells can improve suppressed immune function in patients with ATL, we believe that ANK therapy is much more effective as a first-line therapy than conventional therapies.

In this study, higher NK cell activity was detected in two surviving patients who maintained CR status, demonstrating an improvement in their immune status.

ANK therapy was also reported to be effective in a patient with smoldering ATL with HTLV-1-related bronchoalveolar disorder (HABA) and to improve respiratory function [31]. In this reported HABA patient, chronic lung inflammation appeared to be associated with and regulated by ATL cells, which produce proinflammatory cytokines. These results suggest that ANK therapy has a strong impact on both ATL tumor stem cells and HTLV-1-associated inflammation; thus, it may become the leading treatment modality for ATL.

## 4. Conclusions

Overall, ANK therapy is relatively safe for elderly patients because it has fewer side effects. It is now clear that NK cells remember previous activation events with distinct forms of innate memory and memory-like responses following hapten exposure, viral infection, and combined activation with IL-12, IL-15, and IL-18. There are no accurate data on the number of patients diagnosed with ATL, but previous reports have indicated that approximately 1000 people develop the disease annually in Japan. Thus, although the number of patients is small, our experience of inducing CR status and long-term survival in patients with ATL in this study, along with reports of the efficacy of memory-like NK cell therapy in AML patients, suggest that ANK therapy may become the main treatment for ATL and ATL-related inflammatory diseases in the future. Further efforts are needed to determine which malignant tumors ANK therapy is effective for through therapeutic application and long-term follow-up observation.

## Figures and Tables

**Figure 1 reports-07-00080-f001:**
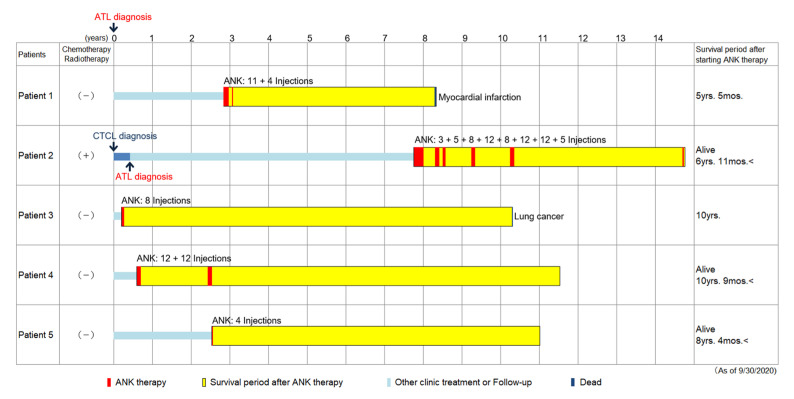
ANK therapy and overall survival. Light blue bars: time from clinical diagnosis to initiation of ANK therapy. Yellow bars: time from initiation of ANK therapy to death or last follow-up day (September 2020). Red bars: duration of ANK therapy. Small numbers: number of intravenous injections of ANK cells.

**Figure 2 reports-07-00080-f002:**
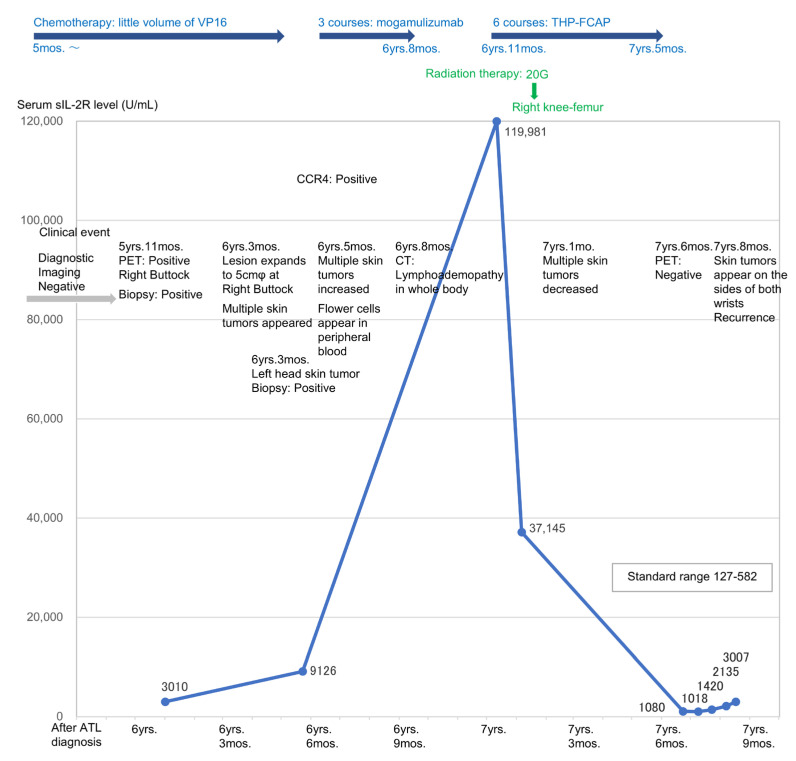
Fluctuation of serum sIL-2R levels in Patient 2 (blue broken line). Serum levels of sIL-2R before ANK therapy. Patient 2 developed acute exacerbation and was treated with multi-agent chemotherapy. She had CR, and her serum sIL-2R levels decreased to 1018 U/mL. However, her ATL relapsed two months later.

**Figure 3 reports-07-00080-f003:**
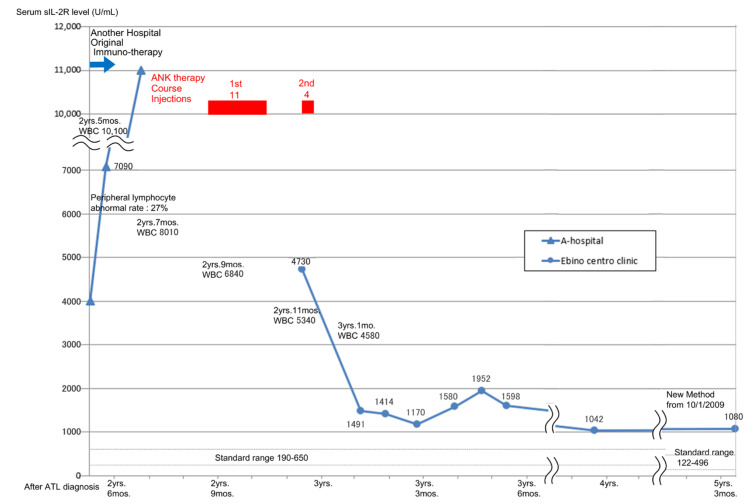
Fluctuation of serum sIL-2R levels in Patient 1 with ANK therapy after acute exacerbation of smoldering ATL. Reprinted/adapted with permission from Ref. [13]. Copyright 2018, copyright Dr. Keisuke Teshigawara.

**Figure 4 reports-07-00080-f004:**
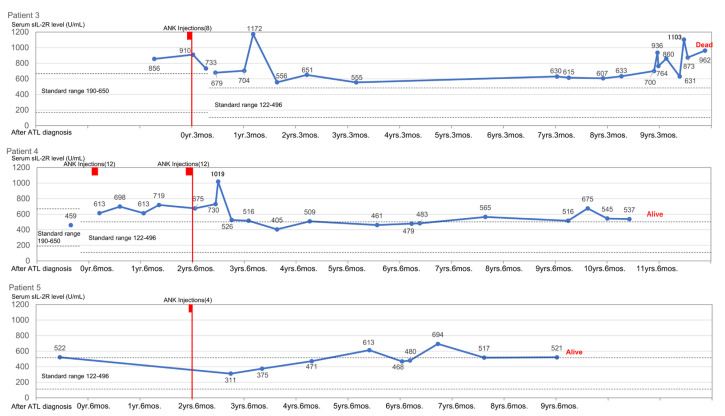
The level of serum sIL-2R in patients receiving ANK therapy as first-line treatment. Serum sIL-2R levels in the patients who received ANK therapy as first-line treatment and their status of watchful waiting.

**Figure 5 reports-07-00080-f005:**
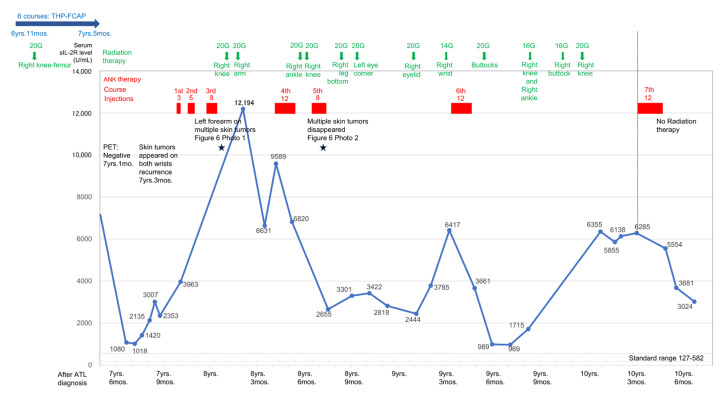
Changes in serum sIL-2R levels in patient 2. Two months after the introduction of ANK therapy, ATL recurred and skin tumors appeared frequently. Radiotherapy was administered until the seventh course of ANK therapy, but then ANK therapy alone was administered for the next 9 months, and serum sIL-2R levels decreased.

**Figure 6 reports-07-00080-f006:**
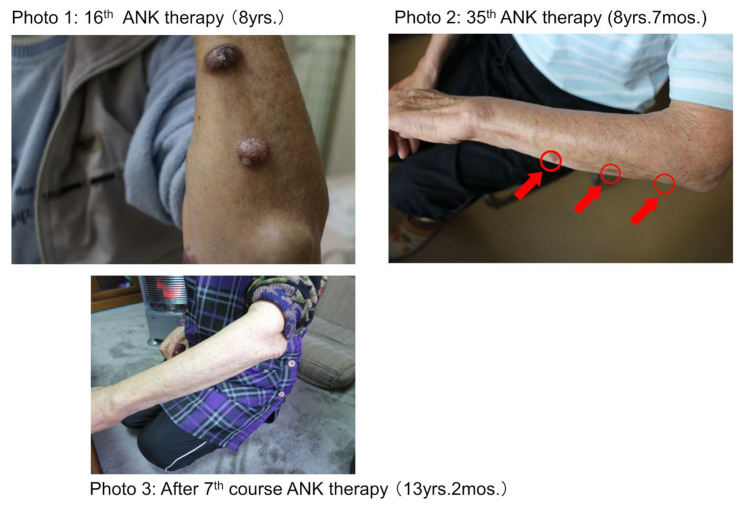
The effect of conventional and ANK therapies on the clinical course of Patient 2. Healing of multiple skin tumor and reduction in serum sIL-2R in Patient 2. Patient 2 relapsed and showed multiple skin tumors on her left forearm (Photo 1). These skin tumors were almost cured seven months after ANK therapy (Photo 2). Thereafter, the healthy status of the left forearm continued for five years (Photo 3).

**Figure 7 reports-07-00080-f007:**
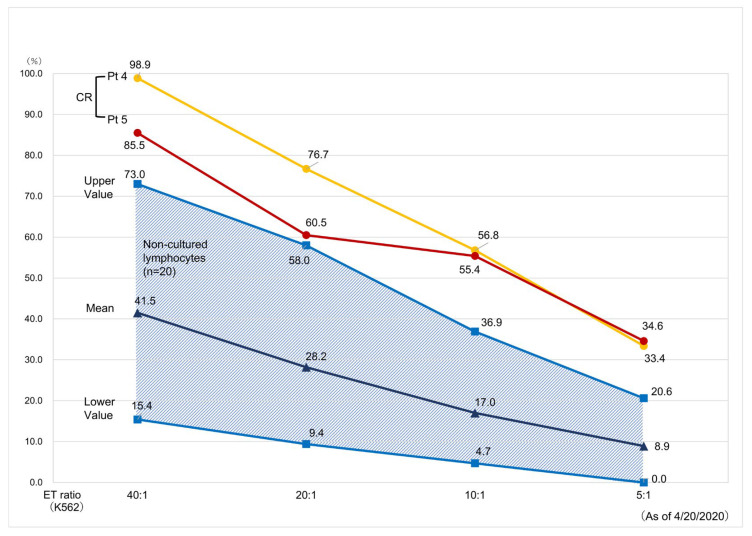
Comparison of NK cell activity between patents achieved complete remission and healthy controls (*n* = 20). Yellow line: Patient 4, red line: Patient 5.

**Table 1 reports-07-00080-t001:** Patient profile and classification according to risk factors [7,8,9]. ANK therapy was administered as first-line therapy to Patients 1, 3, 4 and 5, and as second-line therapy to Patient 2 at the time of relapse after induction of CR with multiagent chemotherapy.

Patient	Age Gender (Age when ANK Therapy was Initiated)	Chemotherapy Radiation Therapy	ATL Diagnosis and Type	Risk Group	Skin Lesions		Evaluation at Treatment
					Before ANK Therapy	After ANK Therapy	
1	75F	(-)	Smoldering	High-risk	Multiple skin erythemas	(-)	CR
			Indolent				
2	71F	(+)	Lymphoma	High-risk	Multiple skin tumors	(-)	PR
			Aggressive				
3	77M	(-)	Lymphoma	Low-risk	Rash	(-)	CR
			Aggressive				
4	70F	(-)	Lymphoma	Low-risk	Rash	(-)	CR
			Aggressive				
5	63F	(-)	Smoldering	Low-risk	Multiple skin erythemas	(-)	CR
			Indolent				

## Data Availability

The original data presented in this study are available on reasonable request from the corresponding author. The data are not publicly available due to privacy.

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
