# Peer review of "Long-Term Survival of Patients with Adult T-Cell Leukemia/Lymphoma Treated with Amplified Natural Killer Cell Therapy"

_reports, 2024, doi:10.3390/reports7030080_

Round 1
Reviewer 1 Report
Comments and Suggestions for Authors
The article is devoted to biotherapeutic option for treatment of adult T-cell leukemia (ATL) associated with HTL virus type 1 (HTLV-1) which is, generally, a drug-resistant lymphoid malignancy affecting elderly persons. An original therapy with amplified natural killer cells (ANK) was used as the first-line treatment in 5 patients followed by longitudinal observation. ANK therapy proceeded in 1-2 courses (except of one patient subjected to multi-agent therapy). Long-term relapse- or progression-free survival was documented in 4 out of 5 cases treated with ANKs.
Specific remarks.
Description of cases:
Line 82: autologous blood cells should be mentioned as a source of ANK.
Line 90: Thein authors must briefly describe the test system used for evaluation of NK activities in final cell product (e.g., which target cells were used?).
Line 91: Please present data on viability and NK activity of thawed and recultured ANKs.
In case description (Lines 99-100, Table 1): one should refer to the common staging/risk assessment system used in this study.
Lines 98-105: Laboratory data lack the results of skin tumor histology in the patients and HTLV-1 status of ANK-treated patients.
Line 149: Please describe in more details any adverse effects in the patients 1 and 3-5 after ANK injections using CTCAE or other common criteria.
Discussion
There are no data on survival in comparison group of ATL patients subjected to conventional therapies. Please present own or literature
Minor remarks
Line 25: “was shown” is better than “revealed”.
Line 123: “experienced” may be replaced by “achieved” or “developed”
Line 125: “lesions” instead of “legions”
Comments on the Quality of English LanguageModerate copy-editing required
Author Response
Thank you very much for taking the time to review our manuscript. Please find the detailed responses below and the corresponding revisions in track changes in the re-submitted manuscript.

Reviewer 2 Report
Comments and Suggestions for Authors
In the submitted manuscript the authors reported a long-lasting complete remission in four patients with Adult T-cell leukemia/lymphoma (ATL) and cutaneous manifestations after the administration of Amplified-Natural-Killer Cell (ANK) therapy as first-line treatment. Clinical data are well-described and the dynamics of s-IL2R levels are graphically represented.
There are several issues that need to be considered:
- Introduction: In the phrase “ATL is classified into four clinical types: lymphoma, chronic, and smoldering” the acute form is missing
- Please, reformulate the phrase “Recently, it has been reported that soluble interleukin-2 receptor (sIL-2R) is an important poor prognostic factor”, such as “elevated serum levels of sIL-2R represent an adverse prognostic factor…
- The statement “The mechanism underlying the effectiveness of ANK therapy is considered that ANK therapy induces memory NK cells to maintain high NK cell activity, thereby killing remaining ATL cells after CR or prevents HTLV-1 infected cells from transformation” lacks a reference and needs also reformulation in terms of English
- Please. describe briefly the method of NK cell activity assessment
- In the Discussion section we recommend to insert a short paragraph related to the current status of NK cell therapy in oncology
- Conclusions: The statement “Furthermore, although there is no direct evidence, our study outcomes indicate that ANK therapy may have a strong effect on preventing the development of ATL, including leukemia stem cells” is not clear. Please, reformulate it.
Comments on the Quality of English LanguageThe text requires a revision in terms of English language: e.g. “4 years survival rates” (4-year survival rates); the phrases “One skin tumor appeared on the right buttock at 5 years 11 months after the diagnosis and 6 years and 3 months later, multiple skin tumors appeared and 7 years later, sIL-2R rose to 119,981 106 U/mL, diagnosed with acute exacerbation of ATL”, “These skin lesions are induced by ATL producing proinflammatory cytokines and/or ATL invasion. Especially, the skin legions of nodulotumoral and erythrodermic types are derived from ATL invasion”, “Due to the limited number of ANK therapy compared to ATL cells proliferating at the time of recurrence, the number of ANK cells might not be sufficient to kill all tumor cells”, “IL-10 and IL-17 suppress the enhanced immune response induced by IFN-γ is associated with skin inflammation”, should be corrected/reformulated.
Author Response
Thank you very much for taking the time to review our manuscript. Please find the detailed responses below and the corresponding revisions/corrections highlighted in track changes in the re-submitted manuscript.

Round 2
Reviewer 1 Report
Comments and Suggestions for Authors
In the revised version of article on NK therapy in ATLL patients, a sufficient work was performed in order to make the text more convincing and understandable. Multiple misprints are corrected, and some ambigous points are specified, e.g., source of therapeutic NK oreoarations and their biological testing (lines 84-90).
In Discussion, the authors referred to previous studies on usage of NK cells for ATLL treatment which seem to show more clinical effect than currently used chemotherapy. Moreover, NK-cell therapy may be more effective than T-cell biotherapy (line 215). However, these promising results should be confirmed in more extensive clinical trials within cooperative studies, due to relative low incidence of ATLL which could be mentioned in Discussion and Conclusions as a limitation of current study.
Comments on the Quality of English LanguageMinor editing of the text is required
Author Response
Thank you very much for taking the time to review our manuscript. Please find the detailed responses below and the corresponding revisions/corrections highlighted/in track changes in the re-submitted manuscript.
